# Bis-thiobarbiturates as Promising Xanthine Oxidase Inhibitors: Synthesis and Biological Evaluation

**DOI:** 10.3390/biomedicines9101443

**Published:** 2021-10-11

**Authors:** João L. Serrano, Diana Lopes, Melani J. A. Reis, Renato E. F. Boto, Samuel Silvestre, Paulo Almeida

**Affiliations:** 1CICS-UBI—Health Sciences Research Center, University of Beira Interior, Av. Infante D. Henrique, 6200-506 Covilhã, Portugal; joao.serrano@ubi.pt (J.L.S.); nekolopes@hotmail.com (D.L.); joanareis_95@hotmail.com (M.J.A.R.); rboto@ubi.pt (R.E.F.B.); 2CNC—Center for Neuroscience and Cell Biology, University of Coimbra, Rua Larga, 3004-517 Coimbra, Portugal

**Keywords:** bis-thiobarbiturates, xanthine oxidase, antioxidant, cytotoxicity, in silico evaluation

## Abstract

Xanthine oxidase (XO) is the enzyme responsible for the conversion of endogenous purines into uric acid. Therefore, this enzyme has been associated with pathological conditions caused by hyperuricemia, such as the disease commonly known as gout. Barbiturates and their congeners thiobarbiturates represent a class of heterocyclic drugs capable of influencing neurotransmission. However, in recent years a very large group of potential pharmaceutical and medicinal applications have been related to their structure. This great diversity of biological activities is directly linked to the enormous opportunities found for chemical change off the back of these findings. With this in mind, sixteen bis-thiobarbiturates were synthesized in moderate to excellent reactional yields, and their antioxidant, anti-proliferative, and XO inhibitory activity were evaluated. In general, all bis-thiobarbiturates present a good antioxidant performance and an excellent ability to inhibit XO at a concentration of 30 µM, eight of them are superior to those observed with the reference drug allopurinol (Allo), nevertheless they were not as effective as febuxostat. The most powerful bis-thiobarbiturate within this set showed in vitro IC_50_ of 1.79 μM, which was about ten-fold better than Allo inhibition, together with suitable low cytotoxicity. In silico molecular properties such as drug-likeness, pharmacokinetics, and toxicity of this promising barbiturate were also analyzed and herein discussed.

## 1. Introduction

Xanthine oxidase (XO) is a molybdoflavoprotein widely disseminated throughout the human body, and is present in the liver, intestine, lungs, kidneys, heart, brain, and plasma [1]. Physiologically, XO catalyzes the oxidative hydroxylation of hypoxanthine and xanthine into uric acid (UA) with the concomitant production of reactive oxygen species (ROS) [1]. Therefore, this enzyme is an important source of superoxide radicals and hydrogen peroxide, which contributes to oxidative stress and takes part in the aging process. In addition, these ROS are involved in several pathological processes such as atherosclerosis and cancer [2,3]. Moreover, the excessive activity of XO leads to the overproduction of UA, increasing its concentration in the bloodstream and can result in hyperuricemia [4]. Hyperuricemia is a predisposing factor of gout, whether by excessive production of UA or under-excretion by the kidneys and is also considered a lifestyle-related syndrome. In fact, it has been associated with a high intake of foods rich in nucleic acids, such as red meats and seafood, leading to excessive production of UA [5,6].

Gout is a form of inflammatory arthritis characterized by the chronic deposition of monosodium urate crystals on the joints [4,6]. Hyperuricemia is a critical factor, not only for the development of gout but also for chronic nephritis, cardiovascular diseases, hypertension, type II diabetes mellitus, and metabolic syndrome [6]. The standard therapy for gout relies on urate-lowering drugs, especially those that target XO [4,6], such as allopurinol (Allo), febuxostat, and topiroxostat [1,6,7,8].

The synthesis and biological evaluation of nitrogen-based heterocyclic compounds has received increasing attention over the years [9]. Barbituric and thiobarbituric acids are examples of these heterocycles. Firstly synthesized by Adolph von Baeyer in 1864, barbituric acid was the key molecule for developing new derivatives with central nervous system (CNS) depressing activity in the twentieth century [10]. Initially used as anticonvulsant, anxiolytic, sedative-hypnotic and anesthetics [10], barbituric and thiobarbituric acid derivatives have recently been associated with several potential industrial and pharmaceutical/biological applications [10,11] namely as antifungal [12], antibacterial [13,14,15], anticancer [11,13,16,17], antiviral [18], as well as urease [19], XO [13], and helicase inhibitors [20]. Additionally, it has been established that this pyrimidine class of compounds undergoes Knoevenagel condensations with aldehydes to give 5-ylidene derivatives [13,21]. Furthermore, the formation of Michael adducts often follows Knoevenagel condensations with the formation of novel bis-thiobarbiturates [21]. On the other hand, bis-thiobarbiturates have been scarcely explored except that they have been established as urease inhibitors [22] and were tested as antibacterials but without any relevant activity [15].

## 2. Materials and Methods

### 2.1. Chemicals and Instrumentation

All reagents and solvents used were analytically pure and were used without purification. Ethanol was purchased from Honeywell (Paris, France), acetic acid from José M. Vaz Pereira, S. A. (Sintra, Portugal) and petroleum ether from Chem-Lab (Zedelgem, Belgium). Barbituric and 2-thiobarbituric acids and 2,4-dinitrobenzaldehyde were obtained from Alfa Aesar (Kandel, Germany), while 2-nitrobenzaldehyde was from Maybridge (Loughborough, United Kingdom). *N*,*N*-Diethyl-2-thiobarbituric acid, 4-formylbenzonitrile, 4-methoxybenzaldehyde, 5-hydroxy-2-nitrobenzaldehyde, 6-nitrobenzo[*d*][1,3]diaxole-5-carbaldehyde and 3-pyridinecarboxaldehyde were acquire from Acros Organics (Geel, Belgium). Benzaldehyde, 4-methylbenzaldehyde, 4-nitrobenzaldehyde, *N*-(4-formylphenyl)acetamide, 4-bromobenzaldehyde, 3-hydroxybenzaldehyde, 4-(dimethylamino)-2-nitrobenzaldehyde and α-methyl-*trans*-cinnamaldehyde were obtained from Sigma-Aldrich (St. Louis, MO, USA).

All reactions were monitored by thin-layer chromatography on precoated silica-gel aluminum plates of 0.2 mm (Macherey-Nagel 60 G/UV254, Düren, Germany). After the elution, the plates’ observation was performed under ultraviolet (UV) light with a wavelength of 254 and/or 365 nm.

The melting points (mp) were determined in open capillary tubes using a Büchi B-540 mp apparatus and were not corrected.

Proton (^1^H NMR) and carbon nuclear magnetic resonance (^13^C NMR) spectra were performed on a Bruker Avance III 400 MHz spectrophotometer and were processed by the software MestReNova 14.2.0 lite (Mestrelab Research S.L., Santiago de Compostela, Spain). Deuterated chloroform (CDCl_3_, Acros Organics, Geel, Belgium) or hexadeuterodimethyl sulfoxide (DMSO-*d*_6_, Eurisotop, Gif-sur-Yvette, France) were used as a solvent and internal standard, δ = 7.26 and 77.16 ppm or 2.50 and 39.52 ppm in ^1^H and ^13^C NMR, respectively. The chemical shift (δ) values are given in parts per million (ppm) and coupling constants (*J*) in Hertz (Hz). The multiplicity of the signals is reported as singlet (s), doublet (d), doublet of doublets (dd), doublet of triplets (dt), doublet of doublet of triplets (ddt), triplet (t), triplet of doublets (td), triplet of triplets (tt), quartet (q), quartet of doublets (qd), or multiplet (m).

High-resolution mass spectrometry (HRMS) was performed for new compounds by electrospray ionization time-of-flight (ESI-TOF) at CACTI services from the University of Vigo (Spain).

### 2.2. Synthesis of Bis-thiobarbiturates ***3**–**19***

A mixture of barbituric acid **1c** or thiobarbituric acids **1a**-**b** (2.0 mmol) and a benzaldehyde **2a**-**m** or aldehydes **2n**-**o** (1.0 mmol) in ethanol (5 mL) was stirred for two to six hours, at room temperature (rt) or by reflux [22] or alternatively in acetic acid (5 mL) at 80 °C [23]. The obtained solid was filtered, washed with cold ethanol and petroleum ether 40–60 °C, dried, and recrystallized from ethanol to afford the following bis-barbiturate and bis-thiobarbiturates **3**–**19**.


**5,5′-(Phenylmethylene)bis(1,3-diethyl-6-hydroxy-2-thioxo-2,3-dihydropyrimidin-4(1*H*)-one)** **(3)**


From *N*,*N*-diethyl-2-thiobarbituric acid (**1a**, 2.0 mmol, 400.6 mg) and benzaldehyde (**2a**, 1.0 mmol, 106.1 mg, 101.6 µL) in ethanol at rt; yellow solid (434.9 mg, 89% yield); mp: 177–179 °C; ^1^H NMR (400 MHz, CDCl_3_) δ (ppm) 7.32 (t, *J* = 7.7 Hz, 2H, 2 × ArCH), 7.26 (t, *J* = 7.4 Hz, 1H, ArCH), 7.13 (d, *J* = 7.9 Hz, 2H, 2 × ArCH), 5.68 (s, 1H, 5-CCH), 4.76–4.50 (m, 8H, 4 × NCH_2_CH_3_), 1.38 (t, *J* = 7.0 Hz, 6H, 2 × NCH_2_CH_3_), 1.29 (t, *J* = 7.0 Hz, 6H, 2 × NCH_2_CH_3_); Appendix A. ^13^C NMR (101 MHz, CDCl_3_) δ (ppm) 174.71 (2 × 2-CS), 163.84 (2 × CO), 162.38 (2 × CO), 135.62 (ArC), 128.58 (2 × ArCH), 126.90 (ArCH), 126.43 (2 × ArCH), 97.55 (2 × 5-C), 45.26 (2 × NCH_2_CH_3_), 44.70 (2 × NCH_2_CH_3_), 35.03 (5-CCH), 12.20 (2 × NCH_2_CH_3_), 12.12 (2 × NCH_2_CH_3_); Appendix A.


**5,5′-(*p*-Tolymethylene)bis(1,3-diethyl-6-hydroxy-2-thioxo-2,3-dihydropyrimidin-4(1*H*)-one)** **(4)**


From *N*,*N*-diethyl-2-thiobarbituric acid (**1a**, 2.0 mmol, 400.6 mg) and 4-methylbenzaldehyde (**2b**, 1.0 mmol, 120.2 mg, 118.0 µL) in ethanol at rt; yellow solid (397.1 mg, 79% yield); mp: 163–164 °C; ^1^H NMR (400 MHz, CDCl_3_) δ (ppm) 7.12 (d, *J* = 8.1 Hz, 2H, 2 × ArCH), 7.00 (d, *J* = 8.3 Hz, 2H, 2 × ArCH), 5.63 (s, 1H, 5-CCH), 4.91–4.47 (m, 8H, 4 × NCH_2_CH_3_), 2.34 (s, 3H, ArCCH_3_), 1.38 (t, *J* = 7.0 Hz, 6H, 2 × NCH_2_CH_3_), 1.29 (t, *J* = 7.0 Hz, 6H, 2 × NCH_2_CH_3_); Appendix A. ^13^C NMR (101 MHz, CDCl_3_) δ (ppm) 174.66 (2 × 2-CS), 163.79 (2 × CO), 162.34 (2 × CO), 136.46 (ArC), 132.44 (ArC), 129.28 (2 × ArCH), 126.29 (2 × ArCH), 97.67 (2 × 5-C), 45.23 (2 × NCH_2_CH_3_), 44.65 (2 × NCH_2_CH_3_), 34.71 (5-CCH), 21.09 (ArCCH_3_), 12.19 (2 × NCH_2_CH_3_), 12.11 (2 × NCH_2_CH_3_); Appendix A.


**4-(Bis(1,3-diethyl-6-hydroxy-4-oxo-2-thioxo-1,2,3,4-tetrtahydropyrimidin-5-yl)methyl)benzonitrile** **(5)**


From *N*,*N*-diethyl-2-thiobarbituric acid (**1a**, 2.0 mmol, 400.6 mg) and 4-formylbenzonitrile (**2c**, 1.0 mmol, 131.1 mg) in ethanol at rt; white solid (400.6 mg, 78% yield); mp: 199–200 °C; ^1^H NMR (400 MHz, CDCl_3_) δ (ppm) 7.62 (d, *J* = 8.5 Hz, 2H, 2 × ArCH), 7.27 (d, *J* = 8.5 Hz, 2H, 2 × ArCH), 5.66 (s, 1H, 5-CCH), 4.90–4.42 (m, 8H, 4 × NCH_2_CH_3_), 1.37 (t, *J* = 7.0 Hz, 6H, 2 × NCH_2_CH_3_), 1.28 (t, *J* = 7.0 Hz, 6H, 2 × NCH_2_CH_3_); Appendix A. ^13^C NMR (101 MHz, CDCl_3_) δ (ppm) 174.71 (2 × 2-CS), 163.93 (2 × CO), 162.35 (2 × CO), 141.76 (ArC), 132.40 (2 × ArCH), 127.42 (2 × ArCH), 118.74 (ArCCN), 110.92 (ArC), 96.73 (2 × 5-C), 45.35 (2 × NCH_2_CH_3_), 44.80 (2 × NCH_2_CH_3_), 35.43 (5-CCH), 12.14 (2 × NCH_2_CH_3_), 12.12 (2 × NCH_2_CH_3_); Appendix A.


**5,5′-((4-Nitrophenyl)methylene)bis(1,3-diethyl-6-hydroxy-2-thioxo-2,3-dihydropirimidin-4(1*H*)-one)** **(6)**


From *N*,*N*-diethyl-2-thiobarbituric acid (**1a**, 2.0 mmol, 400.6 mg) and 4-nitrobenzaldehyde (**2d**, 1.0 mmol, 151.1 mg) in ethanol at rt; yellow solid (416.2 mg, 78% yield); mp: 202–204 °C; ^1^H NMR (400 MHz, CDCl_3_) δ (ppm) 8.18 (d, *J* = 8.9 Hz, 2H, 2 × ArCH), 7.33 (d, *J* = 8.9 Hz, 2H, 2 × ArCH), 5.69 (s, 1H, 5-CCH), 4.77–4.38 (m, 8H, 4 × NCH_2_CH_3_), 1.37 (t, *J* = 7.0 Hz, 6H, 2 × NCH_2_CH_3_), 1.29 (t, *J* = 7.0 Hz, 6H, 2 × NCH_2_CH_3_); Appendix A. ^13^C NMR (101 MHz, CDCl_3_) δ (ppm) 174.72 (2 × 2-CS), 163.93 (2 × CO), 162.39 (2 × CO), 146.92 (ArC), 143.88 (ArC), 127.56 (2 × ArCH), 123.82 (2 × ArCH), 96.82 (2 × 5-C), 45.38 (2 × NCH_2_CH_3_), 44.82 (2 × NCH_2_CH_3_), 35.43 (5-CCH), 12.14 (2 × NCH_2_CH_3_), 12.12 (2 × NCH_2_CH_3_); Appendix A.


**5,5′-((4-Nitrophenyl)methylene)bis(6-hydroxy-2-thioxo-2,3-dihydropyrimidin-4(1*H*)-one)** **(7)**


From 2-thiobarbituric acid (**1b**, 2.0 mmol, 258.6 mg) 4-nitrobenzaldehyde (**2d**) (1.0 mmol, 152.3 mg) in ethanol at reflux; pale yellow solid (303.5 mg; 72% yield); mp: 225 °C dec.; ^1^H NMR (400 MHz, DMSO-*d*_6_) δ (ppm) 11.76 (s, 4H, 4 × NH), 8.07 (d, *J* = 8.7 Hz, 2H, 2 × ArCH), 7.27 (d, *J* = 8.5 Hz, 2H, 2 × ArCH), 6.04 (s, 1H, 5-CCH); Appendix A. ^13^C NMR (101 MHz, DMSO-*d*_6_) δ (ppm) 173.16 (2 × 2-CS), 163.04 (4 × CO), 151.99 (ArC), 145.34 (ArC), 127.88 (2 × ArCH), 123.16 (2 × ArCH), 95.28 (2 × 5-C), 31.29 (5-CCH); Appendix A.


**5,5′-((4-Nitrophenyl)methylene)bis(6-hydroxypyrimidine-2,4(1*H*,3*H*)-dione)** **(8)**


From barbituric acid (**1c**, 2.0 mmol, 231.8 mg) 4-nitrobenzaldehyde (**2d**) (1.0 mmol, 152.3 mg) in ethanol at reflux during six hours; white solid (319.2 mg, 82% yield); mp: 227 °C dec.; ^1^H NMR (400 MHz, DMSO-*d*_6_) δ (ppm) 10.74 (s, 4H, 4 × NH), 8.08 (d, *J* = 8.7 Hz, 2H, 2 × ArCH), 7.37 (d, *J* = 8.4 Hz, 2H, 2 × ArCH), 5.82 (s, 1H, 5-CCH); Appendix A.^13^C NMR spectra was not herein present due to a rapid product decomposition in DMSO-*d*_6_ solution.


***N*-(4-(Bis(1,3-diethyl-6-hydroxy-4-oxo-2-thioxo-1,2,3,4-tetrahydropyrimidin-5-yl)methyl)phenyl)acetamide** **(9)**


From *N*,*N*-diethyl-2-thiobarbituric acid (**1a**, 2.0 mmol, 400.6 mg) and *N*-(4-formylphenyl)acetamide (**2e**, 1.0 mmol, 167.2 mg) in ethanol at rt; yellow solid (485.6 mg, 89% yield); mp: 178–179 °C; ^1^H NMR (400 MHz, CDCl_3_) δ (ppm) 7.47 (d, *J* = 8.3 Hz, 2H, 2 × ArCH), 7.17 (s, 1H, NHCOCH_3_), 7.06 (d, *J* = 8.1 Hz, 2H, 2 × ArCH), 5.62 (s, 1H, 5-CCH), 4.74–4.52 (m, 8H, 4 × NCH_2_CH_3_), 2.17 (s, 3H, NHCOCH_3_), 1.37 (t, *J* = 6.9 Hz, 6H, 2 × NCH_2_CH_3_), 1.28 (t, *J* = 7.0 Hz, 6H, 2 × NCH_2_CH_3_); Appendix A. ^13^C NMR (101 MHz, CDCl_3_) δ (ppm) 174.47 (2 × 2-CS), 168.76 (NHCOCH_3_), 163.59 (2 × CO), 162.13 (2 × CO), 137.25 (ArC), 130.59 (ArC), 126.72 (2 × ArCH), 119.64 (2 × ArCH), 97.40 (2 × 5-C), 45.06 (2 × NCH_2_CH_3_), 44.48 (2 × NCH_2_CH_3_), 34.45 (5-CCH), 24.37 (NHCOCH_3_), 12.01 (2 × NCH_2_CH_3_), 11.95 (2 × NHCH_2_CH_3_); Appendix A. HMRS (ESI-TOF): *m*/*z* [M + H]^+^ calcd for C_25_H_32_N_5_O_5_S_2_: 546.1839; found: 546.1840.


**5,5′-((4-Methoxyphenyl)methylene)bis(1,3-diethyl-6-hydroxy-2-thioxo-2,3-dihydropyrimidin-4(1*H*)-one)** **(10)**


From *N*,*N*-diethyl-2-thiobarbituric acid (**1a**, 2.0 mmol, 400.6 mg) and 4-methoxybenzaldehyde (**2f**, 1.0 mmol, 136.2 mg) in acetic acid at 80 °C; yellow solid (352.7 mg, 68% yield); mp: 139–140 °C; ^1^H NMR (400 MHz, CDCl_3_) δ (ppm) 7.02 (dd, *J* = 8.8, 1.0 Hz, 2H, 2 × ArCH), 6.84 (d, *J* = 8.8 Hz, 2H, 2 × ArCH), 5.62 (s, 1H, 5-CCH), 4.75–4.45 (m, 8H, 4 × NCH_2_CH_3_), 3.80 (s, 3H, OCH_3_), 1.37 (t, *J* = 7.0 Hz, 6H, 2 × NCH_2_CH_3_), 1.29 (t, *J* = 7.0 Hz, 6H, 2 × NCH_2_CH_3_); Appendix A. ^13^C NMR (101 MHz, CDCl_3_) δ (ppm) 174.67 (2 × 2-CS), 163.78 (2 × CO), 162.34 (2 × CO), 158.44 (ArC), 127.54 (2 × ArCH), 127.34 (ArC), 113.95 (2 × ArCH), 97.76 (2 × 5-C), 55.39(OCH_3_), 45.26 (2 × NCH_2_CH_3_), 44.69 (2 × NCH_2_CH_3_), 34.36 (5-CCH), 12.20 (2 × NCH_2_CH_3_), 12.14 (2 × NCH_2_CH_3_); Appendix A.


**5,5′-((4-Bromophenyl)methylene)bis(1,3-diethyl-6-hydroxy-2-thioxo-2,3-dihydropyrimidin-4(1*H*)-one)** **(11)**


From *N*,*N*-diethyl-2-thiobarbituric acid (**1a**, 2.0 mmol, 400.6 mg) and 4-bromobenzaldehyde (**2g**, 1.0 mmol, 185.0 mg) in ethanol at rt; white solid (402.9 mg, 64% yield); mp: 169–170 °C; ^1^H NMR (400 MHz, CDCl_3_) δ (ppm) 7.43 (dt, *J* = 8.6, 2.6, 1.9 Hz, 2H, 2 × ArCH), 7.01 (dt, *J* = 8.6, 2.7, 1.8 Hz, 2H, 2 × ArCH), 5.59 (s, 1H, 5-CCH), 4.81–4.45 (m, 8H, 4 × NCH_2_CH_3_), 1.37 (t, *J* = 7.0 Hz, 6H, 2 × NCH_2_CH_3_), 1.29 (t, *J* = 7.0 Hz, 6H, 2 × NCH_2_CH_3_); Appendix A. ^13^C NMR (101 MHz, CDCl_3_) δ (ppm) 174.71 (2 × 2-CS), 163.87 (2 × CO), 162.33 (2 × CO), 134.91 (ArC), 131.66 (2 × ArCH), 128.33 (2 × ArCH), 120.79 (ArC), 97.19 (2 × 5-C), 45.31 (2 × NCH_2_CH_3_), 44.75 (2 × NCH_2_CH_3_), 34.78 (5-CCH), 12.18 (2 × NCH_2_CH_3_), 12.13 (2 × NCH_2_CH_3_); Appendix A.


**5,5′-((3-Hydroxyphenyl)methylene)bis(1,3-diethyl-6-hydroxy-2-thioxo-2,3-dihydropyrimidin-4(1*H*)-one)** **(12)**


From *N*,*N*-diethyl-2-thiobarbituric acid (**1a**, 2.0 mmol, 400.6 mg) and 3-hydroxybenzaldehyde (**2h**, 1.0 mmol, 123.9 mg) in ethanol at rt; yellow solid (183.9 mg, 73% yield); mp: 191–193 °C; ^1^H NMR (400 MHz, CDCl_3_) δ (ppm) 7.12 (t, *J* = 7.9 Hz, 1H, ArCH), 6.68 (ddt, *J* = 8.0, 2.1, 1.0 Hz, 1H, ArCH), 6.60 (dt, *J* = 8.2, 1.9, 0.9 Hz, 1H, ArCH), 6.58 (dd, *J* = 2.1, 1.0 Hz, 1H, ArCH), 5.57 (s, 1H, 5-CCH), 4.75–4.44 (m, 8H, 4 × NCH_2_CH_3_), 1.34 (t, *J* = 7.0 Hz, 6H, 2 × NCH_2_CH_3_), 1.26 (t, *J* = 7.0 Hz, 6H, 2 × NCH_2_CH_3_); Appendix A. ^13^C NMR (101 MHz, CDCl_3_) δ (ppm) 174.58 (2 × 2-CS), 163.61 (2 × CO), 162.28 (2 × CO), 157.36 (ArC), 137.22 (ArC), 129.35 (ArCH), 117.49 (ArCH), 113.93 (ArCH), 113.77 (ArCH), 97.52 (2 × 5-C), 45.11 (2 × NCH_2_CH_3_), 44.60 (2 × NCH_2_CH_3_), 34.85 (5-CCH), 12.12 (2 × NCH_2_CH_3_), 12.04 (2 × NCH_2_CH_3_); Appendix A.


**5,5′-((2-Nitrophenyl)methylene)bis(1,3-diethyl-6-hydroxy-2-thioxo-2,3-dihydropyrimidin-4(1*H*)-one)** **(13)**


From *N*,*N*-diethyl-2-thiobarbituric acid (**1a**, 2.0 mmol, 400.6 mg) and 2-nitrobenzaldehyde (**2i**, 1.0 mmol, 151.1 mg) in ethanol at rt; rose solid (474.9 mg, 89% yield); mp: 172–173 °C; ^1^H NMR (400 MHz, CDCl_3_) δ (ppm) 7.56 (dd, *J* = 7.8, 1.4 Hz, 1H, ArCH), 7.52 (td, *J* = 7.7, 1.5 Hz, 1H, ArCH), 7.42 (tt, *J* = 7.6, 1.1 Hz, 1H, ArCH), 7.28 (dt, *J* = 7.9, 1.1 Hz, 1H, ArCH), 6.11 (s, 1H, 5-CCH), 4.63 (q, *J* = 7.0 Hz, 4H, 2 × NCH_2_CH_3_), 4.59–4.47 (m, 4H, 2 × NCH_2_CH_3_), 1.36 (t, *J* = 7.0 Hz, 6H, 2 × NCH_2_CH_3_), 1.28 (t, *J* = 7.0 Hz, 6H, 2 × NCH_2_CH_3_); Appendix A. ^13^C NMR (101 MHz, CDCl_3_) δ (ppm) 174.56 (2 × 2-CS), 163.77 (2 × CO), 162.15 (2 × CO), 150.18 (ArC), 131.44 (ArCH), 129.64 (ArCH), 129.42 (ArC), 128.18 (ArCH), 124.24 (ArCH), 96.73 (2 × 5-C), 45.22 (2 × NCH_2_CH_3_), 44.73 (2 × NCH_2_CH_3_), 32.75 (5-CCH), 12.04 (2 × NCH_2_CH_3_), 11.87 (2 × NCH_2_CH_3_); Appendix A.


**5,5′-((4-(Dimethylamino)-2-nitrophenyl)methylene)bis(1,3-diethyl-6-hydroxy-2-thioxo-2,3-dihydropyrimidin-4(1*H*)-one)** **(14)**


From *N*,*N*-diethyl-2-thiobarbituric acid (**1a**, 2.0 mmol, 400.6 mg) and 4-(dimethylamino)-2-nitrobenzaldehyde (**2j**, 1.0 mmol, 200.2 mg) in ethanol at rt; red solid (490.2 mg, 85% yield); mp: 143–145 °C; ^1^H NMR (400 MHz, CDCl_3_) δ (ppm) 7.10 (d, *J* = 8.3 Hz, 1H, ArCH), 7.06–6.96 (m, 2H, 2 × ArCH), 6.01 (s, 1H, 5-CCH), 4.63 (q, *J* = 7.1 Hz, 4H, 2 × NCH_2_CH_3_), 4.57–4.45 (m, 4H, 2 × NCH_2_CH_3_), 3.03 (s, 6H, N(CH_3_)_2_), 1.36 (t, *J* = 6.8 Hz, 6H, 2 × NCH_2_CH_3_), 1.29 (t, *J* = 7.1 Hz, 6H, 2 × NCH_2_CH_3_); Appendix A. ^13^C NMR (101 MHz, CDCl_3_) δ (ppm) 174.54 (2 × 2-CS), 163.71 (2 × CO), 162.12 (2 × CO), 150.87 (ArC), 147.87 (ArC), 130.62 (ArCH), 118.95 (ArC), 116.46 (ArCH), 109.51 (ArCH), 97.00 (2 × 5-C), 45.22 (2 × NCH_2_CH_3_), 44.72 (2 × NCH_2_CH_3_), 41.78 (N(CH_3_)_2_), 32.12 (5-CCH), 12.08 (2 × NCH_2_CH_3_), 11.94 (2 × NCH_2_CH_3_); Appendix A. HMRS (ESI-TOF): *m*/*z* [M + H]^+^ calcd for C_25_H_33_N_6_O_6_S_2_: 577.1898; found: 577.1891.


**5,5′-((2,4-Dinitrophenyl)methylene)bis(1,3-diethyl-6-hydroxy-2-thioxo-2,3-dihydropyrimidin-4(1*H*)-one)** **(15)**


From *N*,*N*-diethyl-2-thiobarbituric acid (**1a**, 2.0 mmol, 400.6 mg) and 2,4-dinitrobenzaldehyde (**2k**, 1.0 mmol, 196.1 mg) in ethanol at rt; yellow solid (434.0 mg, 75% yield); mp: 169–170 °C; ^1^H NMR (400 MHz, CDCl_3_) δ (ppm) 8.41 (d, *J* = 2.3 Hz, 1H, ArCH), 8.37 (dd, *J* = 8.7, 2.4 Hz, 1H, ArCH), 7.52 (dd, *J* = 8.7, 1.3 Hz, 1H, ArCH), 6.11 (s, 1H, 5-CCH), 4.63 (q, *J* = 7.0 Hz, 4H, 2 × NCH_2_CH_3_), 4.59–4.46 (m, 4H, 2 × NCH_2_CH_3_), 1.37 (t, *J* = 7.0 Hz, 6H, 2 × NCH_2_CH_3_), 1.29 (t, *J* = 7.0 Hz, 6H, 2 × NCH_2_CH_3_); Appendix A. ^13^C NMR (101 MHz, CDCl_3_) δ (ppm) 174.57 (2 × 2-CS), 163.85 (2 × CO), 162.21 (2 × CO), 149.93 (ArC), 146.87 (ArC), 137.06 (ArC), 131.25 (ArCH), 125.67 (ArCH), 119.58 (ArCH), 95.99 (2 × 5-C), 45.34 (2 × NCH_2_CH_3_), 44.87 (2 × NCH_2_CH_3_), 33.34 (5-CCH), 12.01 (2 × NCH_2_CH_3_), 11.89 (2 × NCH_2_CH_3_); Appendix A.


**5,5′-((5-Hydroxy-2-nitrophenyl)methylene)bis(1,3-diethyl-6-hydroxy-2-thioxo-2,3-dihydropyrimidin-4(1*H*)-one)** **(16)**


From *N*,*N*-diethyl-2-thiobarbituric acid (**1a**, 2.0 mmol, 400.6 mg) and 5-hydroxy-2-nitrobenzaldehyde (**2l**, 1.0 mmol, 167.1 mg) in ethanol at rt; yellow solid (505.7 mg, 92% yield); mp: 167–169 °C; ^1^H NMR (400 MHz, CDCl_3_) δ (ppm) 7.53 (d, *J* = 8.6 Hz, 1H, ArCH), 6.74 (dd, *J* = 8.6, 2.5 Hz, 1H, ArCH), 6.70 (d, *J* = 2.2 Hz, 1H, ArCH), 6.10 (s, 1H, 5-CCH), 4.58 (q, *J* = 7.0 Hz, 4H, 2 × NCH_2_CH_3_), 4.49 (m, 4H, 2 × NCH_2_CH_3_), 1.31 (t, *J* = 7.0 Hz, 6H, 2 × NCH_2_CH_3_), 1.22 (t, *J* = 6.9 Hz, 6H, 2 × NCH_2_CH_3_); Appendix A. ^13^C NMR (101 MHz, CDCl_3_) δ (ppm) 174.35 (2 × 2-CS), 163.51 (2 × CO), 161.86 (2 × CO), 160.76 (ArC), 141.96 (ArC), 132.45 (ArC), 127.17 (ArCH), 116.99 (ArCH), 113.93 (ArCH), 97.01 (2 × 5-C), 45.04 (2 × NCH_2_CH_3_), 44.60 (2 × NCH_2_CH_3_), 33.01 (5-CCH), 11.96 (2 × NCH_2_CH_3_), 11.76 (2 × NCH_2_CH_3_); Appendix A. HMRS (ESI-TOF): *m*/*z* [M + H]^+^ calcd for C_23_H_28_N_5_O_7_S_2_: 550.1425; found: 550.1417.



**5,5′-((6-Nitrobenzo[*d*][1,3]dioxol-5-yl)methylene)bis(1,3-diethyl-6-hydroxy-2-thioxo-2,3-dihydropyrimidin-4(1*H*)-one) (17)**



From *N*,*N*-diethyl-2-thiobarbituric acid (**1a**, 2.0 mmol, 400.6 mg) and 6-nitrobenzo[*d*][1,3]diaxole-5-carbaldehyde (**2m**, 1.0 mmol, 199.1 mg) in ethanol at rt; yellow solid (429.4 mg, 77% yield); mp: 148–149 °C; ^1^H NMR (400 MHz, CDCl_3_) δ (ppm) 7.11 (s, 1H, ArCH), 6.67 (d, *J* = 1.1 Hz, 1H, ArCH), 6.11 (s, 2H, OCH_2_O), 6.10 (d, *J* = 1.1 Hz, 1H, 5-CCH), 4.68–4.46 (m, 8H, 4 × NCH_2_CH_3_), 1.36 (t, *J* = 7.0 Hz, 6H, 2 × NCH_2_CH_3_), 1.28 (t, *J* = 7.0 Hz, 6H, 2 × NCH_2_CH_3_); Appendix A. ^13^C NMR (101 MHz, CDCl_3_) δ (ppm) 174.50 (2 × 2-CS), 163.79 (2 × CO), 162.06 (2 × CO), 150.44 (ArC), 146.72 (ArC), 144.08 (ArC), 125.75 (ArC), 109.24 (OCH_2_O), 105.84 (ArCH), 103.09 (ArCH), 97.06 (2 × 5-C), 45.24 (2 × NCH_2_CH_3_), 44.73 (2 × NCH_2_CH_3_), 32.83 (5-CCH), 12.03 (2 × NCH_2_CH_3_), 11.88 (2 × NCH_2_CH_3_); Appendix A. HMRS (ESI-TOF): *m*/*z* [M + H]^+^ calcd for C_24_H_28_N_5_O_8_S_2_: 578.1374; found: 578.1381.


**5,5′-(2-Methyl-3-phenylprop-2-ene-1,1-diyl)bis(1,3-diethyl-6-hydroxy-2-thioxo-2,3-dihydropyrimidin-4(1*H*)-one)** **(18)**


From *N*,*N*-diethyl-2-thiobarbituric acid (**1a**, 2.0 mmol, 400.6 mg) and α-methyl-*trans*-cinnamaldehyde (**2n**, 1.0 mmol, 150.5 mg, 145.1 µL) in ethanol at rt; yellow solid (264.3 mg, 50% yield); mp: 145–146 °C; ^1^H NMR (400 MHz, CDCl_3_) δ (ppm) 7.34 (t, *J* = 7.8 Hz, 2H, 2 × ArCH), 7.25–7.19 (m, 3H, 3 × ArCH), 6.27 (s, 1H, 5-CCH), 5.01 (q, *J* = 2.7, 1.4 Hz, 1H, C=CH), 4.77–4.51 (m, 8H, 4 × NCH_2_CH_3_), 1.77 (t, *J* = 1.3 Hz, 3H, CCH_3_), 1.36 (t, *J* = 7.0 Hz, 6H, 2 × NCH_2_CH_3_), 1.31 (t, *J* = 7.0 Hz, 6H, 2 × NCH_2_CH_3_); Appendix A. ^13^C NMR (101 MHz, CDCl_3_) δ (ppm) 174.62 (2 × 2-CS), 163.61 (2 × CO), 162.39 (2 × CO), 138.02 (ArC), 131.24 (C=CH), 129.04 (2 × ArCH), 128.26 (2 × ArCH), 126.72 (ArCH), 126.60 (C=CH), 97.53 (2 × 5-C), 45.26 (2 × NCH_2_CH_3_), 44.72 (2 × NCH_2_CH_3_), 38.27 (5-CCH), 17.19 (CCH_3_), 12.19 (2 × NCH_2_CH_3_), 12.15 (2 × NCH_2_CH_3_); Appendix A. HMRS (ESI-TOF): *m*/*z* [M + H]^+^ calcd for C_26_H_33_N_4_O_4_S_2_: 529.1938; found: 529.1939.


**5,5′-(Pyridin-3-ylmethylene)bis(1,3-diethyl-6-hydroxy-2-thioxo-2,3-dihydropyrimidin-4(1*H*)-one)** **(19)**


From *N*,*N*-diethyl-2-thiobarbituric acid (**1a**, 2.0 mmol, 400.6 mg) and 3-pyridinecarboxaldehyde (**2o**, 1.0 mmol, 109.3 mg, 95.9 µL) in ethanol at rt; yellow solid (264.4 mg, 54% yield); mp: 253–254 °C; ^1^H NMR (400 MHz, DMSO-*d*_6_) δ (ppm) 8.69 (d, *J* = 5.6 Hz, 1H, ArCH), 8.58 (s, 1H, ArCH), 8.27 (d, *J* = 8.2 Hz, 1H, ArCH), 7.92 (dd, *J* = 8.2, 5.5 Hz, 1H, ArCH), 6.43 (s, 1H, 5-CCH), 4.44 (qd, *J* = 13.0, 6.4 Hz, 8H, 4 × NCH_2_CH_3_), 1.17 (t, *J* = 6.8 Hz, 12H, 4 × NCH_2_CH_3_); Appendix A. ^13^C NMR (101 MHz, DMSO-*d*_6_) δ (ppm) 174.65 (2 × 2-CS), 161.13 (4 × CO), 144.71 (ArCH), 143.15 (ArC), 140.14 (ArCH), 139.24 (ArCH), 126.70 (ArCH), 94.24 (2 × 5-C), 43.12 (4 × NCH_2_CH_3_), 32.24 (5-CCH), 12.30 (4 × NCH_2_CH_3_); Appendix A.

### 2.3. In Vitro Studies

XO inhibitory and antioxidant assays were performed in triplicate while antiproliferative assay was conducted in quadruplicate. For each assay, at least two independent experiments were performed. An initial screening at the concentration of 30 μM for all compounds under study was performed in each in vitro study. A second screening at 5 μM was performed for compounds that originated an inhibitory potential higher than 80% for the XO activity at 30 μM. Half-maximal inhibitory concentration (IC_50_) studies were performed for drugs used as a reference and for the most promising bis-thiobarbiturate in XO inhibitory, antioxidant, and cytotoxicity assays.

#### 2.3.1. Solutions Preparation

For in vitro studies, all bis-thiobarbiturates, Allo, febuxostat, Trolox, and 5-fluorouracil (5-FU) were dissolved in DMSO at the concentration of 10 mM. Additionally, a 10 mM xanthine solution was prepared in a 25 mM sodium hydroxide solution. All solutions were kept at a temperature of 4 °C before each experiment. Allo, febuxostat, Trolox, and 5-FU were purchased from Sigma-Aldrich (St. Louis, MO, USA).

#### 2.3.2. XO Inhibitory Assay

The XO inhibitory activity was evaluated by spectrophotometric quantification of uric acid formation [13]. The 50 mM dihydrogen phosphate buffer (pH 7.4) was used to dilute all solutions. For each assay performed, 50 μL of the test solution and 50 μL of the 0.1 U/mL XO (Sigma-Aldrich X4875, St. Louis, MO, USA) solution were added in each well of an Elisa microplate (96 wells) followed by 5 min of incubation at 37 °C. Final concentrations of 50, 25, 10, 5, 1, and 0.5 µM for Allo, 0.5, 0.075, 0.05, 0.025, 0.01, and 0.001 µM for febuxostat and 10, 5, 2.5, 1, 0.5, and 0.1 µM for bis-thiobarbiturate **11** were used for IC_50_ determinations. The reaction started with the addition of 150 μL of a xanthine solution (420 μM). The absorbance was recorded at a wavelength of 295 nm every minute for 10 min. To obtain only the absorbance associated with uric acid, a solution consisting of 50 μL of the test solution, 150 μL of the xanthine solution, and 50 μL of buffer were used as blank. Additionally, the dihydrogen phosphate buffer was used as a negative control and Allo and febuxostat as positive controls. For each compound, the percentage of enzyme inhibition was calculated according to the following formula:% of XO inhibition = [1 − (ABS_sample_ − ABS_blank of sample_)/ABS_negative control_] × 100

#### 2.3.3. Antioxidant Assay

Antioxidant potential was spectrophotometrically evaluated by the 2,2-diphenyl-1-picrylhydrazyl (DPPH, Sigma-Aldrich, St. Louis, MO, USA) method [13]. All solutions and dilutions were prepared in 99.5% ethanol before each experiment. For each assay, 100 μL of the test solution and 100 μL of DPPH solution (0.2 mM) were added to each well of a 96 wells Elisa microplate. Final concentrations of 120, 60, 60, 15, 7.5, 3.75, and 1 µM for Trolox and bis-thiobarbiturate **11** were used for IC_50_ determinations. After 60 min of incubation in dark at rt, the capacity of each compound to reduce DPPH was followed by measuring the absorbance at 517 nm. Ethanol was used as a negative control and Trolox as the positive control. To discount the absorbance of each compound at 517 nm, a blank was performed with 100 μL of each test compound and 100 μL of ethanol. The antioxidant capacity of each sample was calculated according to the following formula:% DPPH scavenging = [1 − (ABS_sample_ − ABS_blank of sample_)/ABS_negative control_] × 100

#### 2.3.4. Cytotoxicity Assay

The cytotoxic potential of the compounds being studied was evaluated by quantifying the extent of the reduction of 3-(4,5-dimethylthiazol-2-yl)-2,5-diphenyltetrazolium bromide (MTT, VWR, Radnor, PA, USA) [24] on tumor cell lines of the colon (Caco-2) and breast (MCF-7) adenocarcinoma and non-tumor human dermal fibroblasts (NHDF). All cell lines were obtained from the American Type Culture Collection (ATCC, Manassas, VA, USA) and were maintained in 75 cm^2^ culture flasks in a humidified air incubator with 5% CO_2_ at 37 °C. NHDF cells have grown in RPMI 1640 medium (Sigma-Aldrich, St. Louis, MO, USA) supplemented with 10% fetal bovine serum (FBS), 10 mM 4-(2-hydroxyethyl)-1-piperazineethanesulfonic acid (HEPES), 2 mM *L*-glutamine, 1 mM sodium pyruvate, and 1% antibiotic/antimycotic (Ab: 10,000 units/mL penicillin G, 100 mg/mL streptomycin and 25 μg/mL amphotericin B). MCF-7 cells were cultured in high-glucose Dulbecco’s modified Eagle medium (DMEM, Sigma-Aldrich, St. Louis, MO, USA) supplemented with 10% FBS, and 1% Ab. The Caco-2 cell line was cultured in a high glucose DMEM supplemented with 20% FBS and 1% of the antibiotic mixture (Sp: 10,000 units/mL penicillin G and 100 mg/mL of streptomycin). For the assay, cells were seeded in 96-well plates (2 × 10^4^ cells/mL) in the culture medium. After 48 h of adherence, cells were treated with the test solutions and incubated for 72 h. Final concentrations of 200, 100, 50, 10, 1, and 0.1 µM for bis-thiobarbiturate **11** were used for IC_50_ determinations. Untreated cells were used as a negative control and 5-FU as the positive control. Following incubation, the medium was removed and replaced with a fresh incomplete culture medium (without FBS and Ab or Sp) and MTT solution [5 mg/mL in phosphate buffer saline (PBS)]. After further incubation for 4 h at 37 °C, the medium with MTT was removed, formazan crystals were dissolved in DMSO, and the absorbance was read at 570 nm. Results were expressed as the relative cell proliferation in comparison with the negative control cells.

#### 2.3.5. Statistics

All in vitro results are expressed as mean values ± standard deviation (SD) of at least two independent determinations. The difference between groups was analyzed for each assay by Student’s *t*-test. The IC_50_ values were calculated by sigmoidal fitting analysis considering a 95% confidence interval.

### 2.4. In Silico Studies

Drug-likeness of bis-thiobarbiturate **11** was verified by the free web tool SwissADME [25]. Absorption, distribution, metabolism, excretion, and toxicity (ADMET) parameters were evaluated at SwissADME [25] and pkCSM [26] web tools.

## 3. Results and Discussion

Bis-diethylthiobarbiturates **3**–**6**, **9**, and **11**–**19** were synthesized by the method previously described [22]; although, and to the best of our knowledge, compounds **9, 14, 16**–**18** have never been described in the literature. These conditions are characterized by a short reaction time and simplicity of reaction conditions, affording the desired products with a high degree of purity even without crystallization. Furthermore, this method was carried out at rt, and the product was easily isolated by filtration, in moderate to excellent reactional yields, from 50 to 92% (Table 1). Bis-diethylthiobarbiturate **10**, bis-thiobarbiturate **7**, and bis-barbiturate **8** were synthesized by modified processes once the conditions used at rt were not successfully in affording the respective arylidene. Thus, bis-diethylthiobarbiturate **10** was synthesized in acetic acid at 80 °C [23], and bis-(thio)barbiturates **7**–**8** were synthesized in ethanol at reflux temperature. The formation of bis-(thio)barbiturates **3**–**19** was confirmed by NMR, since a singlet in the range of 5.58 to 6.43 ppm and a signal from 31.3 to 38.3 ppm in ^1^H and ^13^C NMR spectra, respectively, were obtained for the aryltrisubstituted methyne group. On the other hand, when arylidene derivatives were formed, a singlet at 8 ppm in ^1^H NMR and 153 ppm in ^13^C NMR were observed [13]. Despite the synthesis and isolation of bis-barbiturate **8**, a DMSO-*d*_6_ solution rapid decomposition to the barbituric acid (**1c**) and respective arylidene was observed (Appendix A), in accordance with the literature [27]. Therefore, bis-barbiturate **8** was not used in further biological studies.

The XO inhibitory assay of the synthesized bis-thiobarbiturates was performed by a spectrophotometric method at 295 nm, and commercial febuxostat and Allo were used as positive controls. The study started with a screening at the concentration of 30 µM, and all sixteen bis-thiobarbiturates demonstrated potential as XO inhibitors (Figure 1 and Appendix A), with inhibitions from 14.92 to 95.72% (88.51% for Allo and 95.03% for febuxostat). After this study, an additional screening at 5 µM was performed for the target compounds that presented an inhibition higher than 80% at 30 µM. Thus, the identification of the most promising bis-thiobarbiturates under study for further concentration-response studies was performed.

In order to perform some structure-activity relationships inferences, firstly the influence of *N* substitution in the thiobarbituric acid moiety was analyzed. Results demonstrated that *N* substitution with ethyl group at thiobarbiturate moiety led to an increment in XO inhibitory activity from 20.86 to 92.25% (**7** versus **6**). Taking both results in mind, several bis-diethylthiobarbiturates with different substituent groups mainly at *para* position of phenyl moiety were analyzed in detail. In this context, electron-withdrawing groups appear to intensify the XO inhibitory potential in relation to electron-donating counterparts. In fact, the presence of nitrile, nitro, or bromo groups at *para* position increases the inhibitory activity from 80.15% (unsubstituted **3**) to 86.12, 92.25, and 95.72% at 30 µM, and from 10.42% to 52.92, 56.82, and 80.92% at 5 µM, for **5**, **6**, and **11**, respectively. On the other hand, the addition of a methyl, acetamide or methoxy group at the same position reduces the inhibitory activity to 36.47, 45.61, and 23.53 at 30 µM, for **4**, **9**, and **10** respectively. Besides these effects observed for different groups in the *para* position, a small decrease in the inhibitory activity was observed when the same group was in the *ortho* position. This effect is notoriously noticed for the bis-thiobarbiturates *para*-nitro and *ortho*-nitro substituted pair, where the values of XO inhibitions are 56.82 and 37.42% at 5 µM, for **6** and **13**, respectively.

The analysis of the influence of the presence of di and tri substitutions in phenyl moiety on the percentage of XO inhibition seems not to be straightforward. Indeed, the additional presence of a dimethylamine group at the *para* position or two methylenedioxy groups at the *meta* and *para* positions of the *ortho*-nitro group substituted seems to increase the XO inhibitory activity at 5 µM (**14** and **17** versus **13**). However, the additional presence of a *para*-nitro or *meta* -hydroxyl decreases the XO inhibitory activity (**15** and **16** versus **13**). A deeply notorious reduction of XO inhibitory activity at 30 µM was observed with a double nitro substitution (14.92%; **15**) concerning the related *ortho-* (89.73%; **13**) or *para-* (92.25%; **6**) mono-nitro congener. Additionally, this effect was also noted for the bis-thiobarbiturates **12** (72.25%; 3-hydroxyl) and **13** (89.73%; 2-nitro) when compared with their related bis-thiobarbiturates **16** (61.44%; 2-nitro, 5-hydroxyl).

Finally, the replacement of the phenyl (bis-thiobarbiturate **3**) moiety for 2-methyl-3-phenylprop-2-enyl or pyridin-3-yl showed a promising increment for activity at 5 µM by about six-fold (**3** versus **18** and **19**).

In conclusion, bis-thiobarbiturates **5**, **6**, **11**, **13**, **14**, **17**, **18**, and **19** evidence a percentage of XO inhibition higher than Allo at 5 µM; however, this is still lower than the second positive control, febuxostat. Generally, the most active compounds present electron-withdrawing or halogen substituents on the phenyl ring. Nevertheless, the effect of further electron-withdrawing and/or electron-donating group substitutions must be carefully balanced.

The halogenated bis-thiobarbiturate **11** demonstrated that it was the most promising within this set of tested compounds. Further concentration-response studies for **11** showed the inhibition of XO was activity-dependent on the concentration, presenting a calculated IC_50_ value of 1.79 μM (Table 2 and Appendix A), being approximately ten-fold more active than the positive control Allo (IC_50_ of 10.73 μM), one of the reference drugs used for gout treatment.

Since both steps of XO purine catabolism led to ROS generation (Appendix A), a dual effect as an XO inhibitor and antioxidant can be profitable for new antigout drugs, since ROS generated by XO can produce cytotoxic effects in many circumstances and thus can promote mutagenesis and tumor development [3]. Therefore, molecules capable of this dual XO inhibitory and radical scavenging activity could be even more advantageous for gout treatment.

The antioxidant potential of all synthesized bis-thiobarbiturates **3**–**7** and **9**–**19** was evaluated by the DPPH method at a concentration of 30 µM, and results were compared with a reference compound, Trolox. The analysis of results (Figure 2 and Appendix A) showed the high antioxidant potential of tested **3**–**7** and **9**–**18**. The only exception was **19** that shows a low DPPH scavenging activity. Withal, the weak antioxidant activity of Allo was also evidenced, as expected [14]. Although a modest influence of several substituents on DPPH scavenging activity at 30 µM was noticed, some interesting conclusions can be inferred. Aligned with XO inhibitory results, the most DPPH scavenging bis-thiobarbiturate is once again the halogenated derivative **11.** In fact, the IC_50_ value determined for **11** is similar to that obtained for Trolox (24.67 and 23.82 µM, respectively) in DPPH radical scavenging activity concentration-response studies (Table 2 and Appendix A).

Regarding the promising results of the bis-thiobarbiturate **11** on both antioxidant and XO inhibitory activity, the in vitro biosafety effectiveness was further analyzed by the MTT method in a non-tumoral cell line. A calculated IC_50_ of 93.15 µM (Table 2 and Appendix A) clearly demonstrated the low cytotoxicity of this compound on NHDF cells at concentrations where XO inhibitory activity was effective. Thus, despite being used in vitro data, it was possible to calculate the selectivity index for bis-thiobarbiturate **11**, with calculated values of 52.04 and 3.77 for XO inhibition and DPPH scavenging activity, respectively.

Although there is no direct relationship between the use of XO inhibitors and a good prognosis in cancer treatment, the expression and activity of this enzyme have been negatively associated with a high degree of malignancy and a worse prognosis in some types of cancer, namely of the breast and gastrointestinal tract, in recent years [28,29]. Considering the low cytotoxicity on NHDF cells observed for bis-thiobarbiturate **11**, the interest of barbiturate and thiobarbiturate derivatives as anticancer agents [11,13] and the recent work on XO inhibitors with anticancer activity [30], we decided to evaluate the cytotoxicity on NHDF as well as the antiproliferative effects on colorectal adenocarcinoma Caco-2 and breast cancer MCF-7 cell lines for all bis-thiobarbiturates under study. To get a strengthened term of comparison, the anticancer drug 5-FU was used as a positive control. From the results (Figure 3 and Appendix A), the not-marked cytotoxicity of almost all tested compounds was noted. In fact, only bis-thiobarbiturates **10**, **13**, **16**, and **17** demonstrated a relevant effect on NHDF cells, with cell viability below 80%. On the other hand, all of the tested bis-thiobarbiturates did not present antiproliferative effects on the breast cancer MCF-7 cell line. Nonetheless, on the colorectal Caco-2 cancer cell line, bis-thiobarbiturates **5**, **9**, **10**, and **11** showed a moderate effect on their cell viability; however, it was weaker than the positive control 5-FU. In addition, compounds **5**, **9**, and **11** demonstrated some selectivity for cancer Caco-2 *versus* normal NHDF cells.

In silico approaches remain a critical tool for drug discovery since their use can be a determinant for a cost-effective identification of promising drug candidates and to reduce the use of animal models [31,32]. In this context, the most promising *bis*-thiobarbiturate herein tested, **11**, was in silico assessed to verify its drug-likeness characteristics with the applicability of Lipinski’s rule of five [33] and Veber’s parameters [34] by the free web tool SwissADME [25]. These two rules are essential tools in drug discovery to predict the potential bioavailability of new drug candidates. Therefore, it was determined that compound **11** respect Lipinski’s rule parameters for hydrogen bond donors (nOHNH), hydrogen bond acceptors (nON) and octanol-water partition coefficient (logP) (Table 3). The exception to this rule was only a molecular weight larger than 500. However, this is not a preponderant factor, since it has been considered that one violation of this rule is acceptable [33]. The alternative analysis of the variant of Lipinski’s rule of five by Veber et al. even claims that a molecular weight cutoff at 500 by itself does not suggest compounds with low bioavailability. Veber’s parameters defend a simple analysis of the number of rotatable bonds (n-rot; 10 or fewer) and topological polar surface area (TPSA; equal to or less than 140 Å^2^ or 12 or fewer nON and nOHNH) [34]. Taking this consideration in mind leads us to conclude that *bis*-thiobarbiturate **11** has a high probability of presenting good bioavailability. Despite presenting TPSA superior to 140 Å^2^, *bis*-thiobarbiturate **11** does not exceed the allowed number of nON and nOHNH.

The identification of **11** as a potential promiscuous compound with reactivity on several biological targets [35] was also crucial for their potential interest in medicinal chemistry. The results obtained eliminate the potential of **11** to be a pan-assay interference compound (PAINS), since no significate alerts were observed. Although some barbituric and thiobarbituric acid derivatives are considered to be highly reactive with numerous biological targets [35], the hypothesis of the *bis*-thiobarbiturate **11** as PAINS is thus removed.

The ADMET parameters of the bis-thiobarbituric **11** were then evaluated on SwissADME [25] and pkCSM [26] web tools. As expected from the fulfillment of Lipinski’s and Veber’s rules, **11** presents potentially good intestinal absorption (71.66%) with moderate solubility in water (Table 4). In addition, compound **11** should not be a potential substrate for P-glycoprotein, an important parameter to consider in the study of drug-drug interactions [36]. Besides, interactions with organic cation transporter 2 (OCT2) [36] and cytochrome P450 (CYP) enzymes [37] are two other important checks to take in mind. In accordance and as expected, **11** reveals not to be a substrate for renal OCT2 CYP1A2, CYP2C19, and CYP2D6 inhibitors. On the other hand, **11** could likely be a potential inhibitor for CYP2C9 and CYP3A4.

As barbiturate derivatives are potential CNS depressants [10], **11**′s distribution profile is additionally analyzed. In this context, in silico SwissADME predictions showed no expectable ability for **11** to permeate the blood-brain barrier (BBB). This low potential capacity of **11** to reach the CNS is corroborated by pkCSM calculated values of log BBB and log CNS.

Besides **11**′s low cytotoxicity on NHDF cells previously established, predictions by pkCSM indicated that their manipulation should not cause skin sensitization. This *bis*-thiobarbiturate also would not be genotoxic or cardiotoxic since it does not have the potential to originate human ether-a-go-go-related gene’s (hERG’s) inhibition. Nonetheless, **11** can be a potential hepatotoxic compound and future studies will be necessary to evaluate their in vitro and/or in vivo toxicity profile.

## 4. Conclusions

Bis-barbiturates and bis-thiobarbiturates were easily and straightforwardly synthesized in moderate to excellent reactional yields. These bis-thiobarbiturates stood out by their antioxidant performance and excellent ability to inhibit the XO at a concentration of 30 µM. The most powerful bis-diethylthiobarbiturate within this set showed an XO inhibition IC_50_ of 1.79 μM, which was about ten-fold better than in vitro Allo inhibition, together with high DPPH radical scavenging activity and suitable low cytotoxicity. The in silico molecular properties such as druglikeness, pharmacokinetics and toxicity were fulfilled for this promising barbiturate, clearly pointing to the potential use of this class of molecules for the treatment of hyperuricemia diseases, such as gout.

## 5. Patents

“Bis-pirimidinonas como inibidores da xantina oxidase para o tratamento de condições patológicas causadas por hiperuricemia” PT116062 (20 January 2020).

## Figures and Tables

**Figure 1 biomedicines-09-01443-f001:**
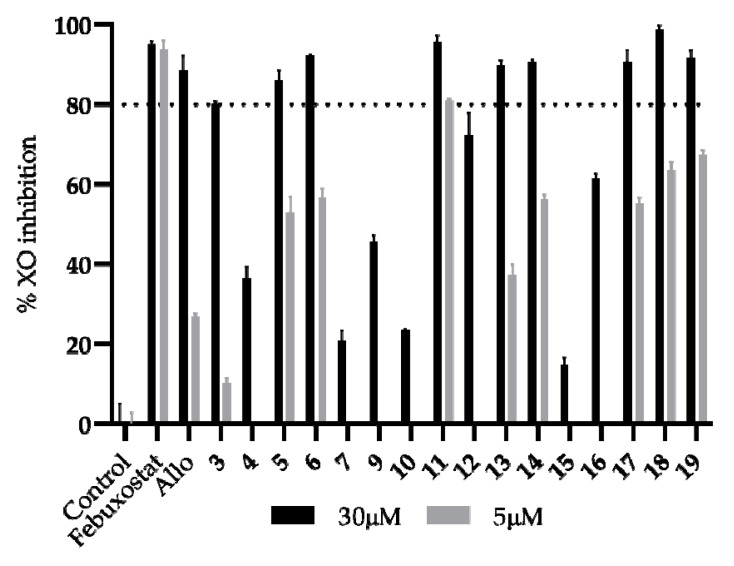
In vitro XO inhibitory activity of bis-thiobarbiturates **3**–**7** and **9**–**19** and references febuxostat and Allo. Results are expressed as average values ± SD of two independent determinations, and each one was performed in triplicate. A *p* < 0.01 versus the negative control in the statistical significance analysis (Student’s *t*-test) was observed for all compounds.

**Figure 2 biomedicines-09-01443-f002:**
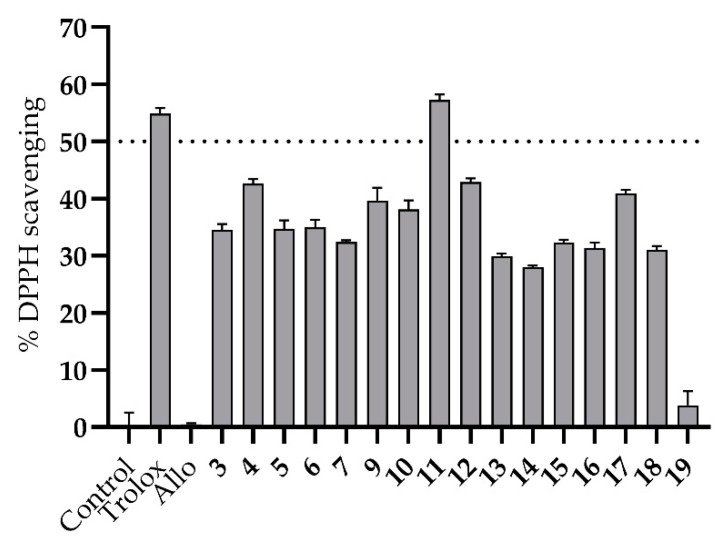
In vitro DPPH radical scavenging activity of bis-thiobarbiturates **3**–**7** and **9**–**19**, Allo and reference Trolox. Results are expressed as average values ± SD of two independent determinations, and each one was performed in triplicated. A *p* < 0.001 versus the negative control in the statistical significance analysis (Student’s *t*-test) was observed for all compounds except for Allo and **19**.

**Figure 3 biomedicines-09-01443-f003:**
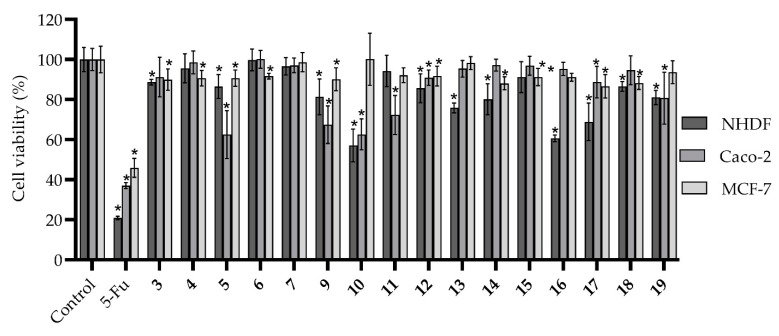
In vitro effects of bis-thiobarbiturates 3–7 and 9–19 and 5-FU on cell viability of non-tumor cell lines of normal human dermal fibroblasts (NHDF), and human adenocarcinoma tumor cell lines of colorectal (Caco-2) and breast (MCF-7), at the single concentration of 30 µM. Results are expressed as average values ± standard deviation of at least two independent assays performed in quadruplicate. * *p* < 0.05 versus the negative control by Student’s *t*-test.

**Table 1 biomedicines-09-01443-t001:** Chemical synthesis, structure and reactional yields of bis-(thio)barbiturates **3**–**19**.

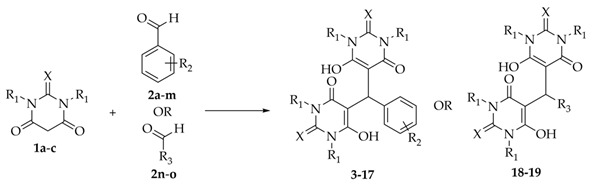
Bis-(thio) Barbiturate	Starting Material	X	R_1_	R_2_	R_3_	Yield (%)
**3**	**1a + 2a**	S	Et	H	-	89
**4**	**1a + 2b**	S	Et	4-CH_3_	-	79
**5**	**1a + 2c**	S	Et	4-CN	-	78
**6**	**1a + 2d**	S	Et	4-NO_2_	-	78
**7**	**1b + 2d**	S	H	4-NO_2_	-	72
**8**	**1c + 2d**	O	H	4-NO_2_	-	82
**9**	**1a + 2e**	S	Et	4-NHCOCH_3_	-	89
**10**	**1a + 2f**	S	Et	4-OCH_3_	-	68
**11**	**1a + 2g**	S	Et	4-Br	-	71
**12**	**1a + 2h**	S	Et	3-OH	-	73
**13**	**1a + 2i**	S	Et	2-NO_2_	-	78
**14**	**1a + 2j**	S	Et	2-NO_2_, 4-N(CH_3_)_2_	-	85
**15**	**1a + 2k**	S	Et	2,4-NO_2_	-	75
**16**	**1a + 2l**	S	Et	2-NO_2_, 5-OH	-	92
**17**	**1a + 2m**	S	Et	2-NO_2_, 4,5-OCH_2_O	-	77
**18**	**1a + 2n**	S	Et	-	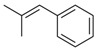	50
**19**	**1a + 2o**	S	Et	-	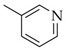	54

**Table 2 biomedicines-09-01443-t002:** In vitro IC_50_ values (µM) for XO inhibition, DPPH radical scavenging activity and cytotoxicity on NHDF cell line of bis-thiobarbiturate **11** and respective references Allo and Trolox ^a^.

	XO Inhibition		DPPH Scavenging		Cytotoxicity on NHDF	
	IC_50_	R^2^	IC_50_	R^2^	IC_50_	R^2^
Febuxostat	0.03 ± 0.01	0.9732	n.d. ^b^	-	n.d. ^b^	-
Allo	10.73 ± 0.81	0.9959	n.d. ^b^	-	n.d. ^b^	-
Trolox	n.d. ^b^	-	23.82 ± 2.13	0.9947	n.d. ^b^	-
**11**	1.79 ± 0.07	0.9986	24.67 ± 0.88	0.9992	93.15 ± 5.54	0.8913

^a^ IC_50_ value ± SD represents as mean at least two independent determinations. ^b^ n.d. is not determined.

**Table 3 biomedicines-09-01443-t003:** In silico molecular properties of bis-thiobarbiturate 11 using the SwissADME predictive database ^a^.

Descriptor		Value
Molecular weight		567.52 g/mol
Log P		4.01
nON		6
nOHNH		2
n-rot		7
TPSA		158.50 Å^2^
Drug-likeness	Lipinski	Yes; 1 violation: MW > 500
	Veber	Yes
Medicinal Chemistry	PAINS	0 alert

^a^ Octanol-water partition coefficient (logP); number of hydrogen bond acceptors (nON); number of hydrogen bond donors (nOHNH); number of rotatable bonds (n-rot); topological polar surface area (TPSA); Lipinski’s rule of five: molecular weight < 500 Da; logP < 5; n-OHNH < 5; n-OHNH < 10. A maximum of 1 violation is permitted [33]. Veber’s parameters: n-rot ≤ 10; TPSA ≤ 140 Å^2^ or total of nON and nOHNH ≤ 12 [34].

**Table 4 biomedicines-09-01443-t004:** In silico pharmacokinetic (absorption, distribution, metabolism, and excretion) and toxicity parameters of bis-thiobarbiturate **11** using pkCSM and SwissADME predictive databases ^a^.

Property	Model Name	Predicted Value
Absorption	Water solubility	−4.66 ^b^ (Moderately soluble)
	Intestinal absorption (human)	71.66%
	P-glycoprotein substrate	No
Distribution	BBB permeant	No
	LogBB	−1.38
	LogPS	−2.42
Metabolism	CYP1A2 inhibitor	No
	CYP2C19 inhibitor	No
	CYP2C9 inhibitor	Yes
	CYP2D6 inhibitor	No
	CYP3A4 inhibitor	Yes
Excretion	Log total clearance	−0.36 ^c^
	Renal OCT2 substrate	No
Toxicity	AMES toxicity	No
	hERG I inhibitor	No
	hERG II inhibitor	No
	Hepatotoxicity	Yes
	Skin Sensitisation	No

^a^ Blood-brain barrier (BBB); logarithm of permeability in blood-brain barrier (logBB) < −1 are poorly distributed; blood-brain permeability-surface area product (logPS) > −2 are considered to penetrate the central nervous system and logPS < −3 are considered as unable to penetrate the central nervous system [26]; human ether-a-go-go-related gene (hERG). ^b^ log(mol/L). ^c^ log(mL/min/kg).

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
