# Peer review of "Bis-thiobarbiturates as Promising Xanthine Oxidase Inhibitors: Synthesis and Biological Evaluation"

_biomedicines, 2021, doi:10.3390/biomedicines9101443_

Round 1

Reviewer 1 Report

The paper submitted by Paulo Almeida and colleagues focuses on the synthesis of derivatives of barbiturates and the study of these derivatives as inhibitors of xanthine oxidase. Although, in my opinion, the manuscript is not ready for publication yet as it needs further improvements for meeting the standards of the Journal. I report some comments below.

1) The scheme of reaction needs to be improved to allow the reader to better understand the mechanisms of the proposed synthesis.

2) In Materials and Methods section it is not well specified which dilutions of compound 11 were made to determine the IC50. It would be better to add the obtained the graphs obtained in the supporting info as well.

3) For compounds, since they have been synthesized, they should be characterized, in addition to NMR, at least also with mass spectrometry to determine the correct MW, and also with liquid chromatography to determine the purity. Furthermore, in SI the NMR spectra have a low quality. Would it be possible to improve it?

4) In the results section the authors would show the negative control data in the bar graphs as well

5) With regard to the in silico data, the study of authors is limited to pharmacokinetic aspects which, moreover, do not demonstrate that the samples are orally active as instead erroneously reported on page 12 line 25, but only provide information about bioavailability. The authors could do in silico studies of docking, which could improve the quality of the paper, given the existence of the crystal in the PDB database; otherwise, in my opinion, putting in silico studies in the title is misleading and therefore in my opinion it should be eliminated.

Author Response

Response to Reviewer #1 Comments

Manuscript ID biomedicines-1400551

Title: Bis-thiobarbiturates as Promising Xanthine Oxidase Inhibitors: Synthesis and Biological Evaluation

Dear Reviewer #1

The authors are extremely grateful and appreciative for the careful revision and very constructive comments made to the Biomedicines submitted manuscript and they would like to thanks for the opportunity to correct and enhance it. All the recommendations were taken into account and changes were made accordingly. Noteworthy main alterations and additions to the manuscript made in the manuscript are marked as word corrections. The responses to the questions and commentaries are herein answered. We hope that this final revised manuscript will be suitable for publication.

With kind regards, looking forward to your decision,

Paulo Almeida

The paper submitted by Paulo Almeida and colleagues focuses on the synthesis of derivatives of barbiturates and the study of these derivatives as inhibitors of xanthine oxidase. Although, in my opinion, the manuscript is not ready for publication yet as it needs further improvements for meeting the standards of the Journal. I report some comments below.

The authors are very pleased and acknowledge the careful revision and the comments.

  • The scheme of reaction needs to be improved to allow the reader to better understand the mechanisms of the proposed synthesis.

Thank you very much for this recommendation. The quality of the simplified scheme in table 1 was improved and the reactional mechanism for the formation of bis-thiobarbiturates was added in Supplementary Information. We hope to meet this goal with the addition of Scheme S1.

  • In Materials and Methods section it is not well specified which dilutions of compound 11 were made to determine the IC50. It would be better to add the obtained the graphs obtained in the supporting info as well.

All the concentrations for the IC50 determinations were added in “Materials and Methods” section. Additionally, all IC50 curves were also added in Supplementary Information as Figures S1-S3. The authors are grateful for this suggestion that undoubtedly led to an improvement of the paper quality.

  • For compounds, since they have been synthesized, they should be characterized, in addition to NMR, at least also with mass spectrometry to determine the correct MW, and also with liquid chromatography to determine the purity. Furthermore, in SI the NMR spectra have a low quality. Would it be possible to improve it?

The authors thanks these recommendations. The quality of all NMR spectra in SI was accordingly improved. Taking in consideration that NMR data of the compounds already described in the literature were as expected, mass spectrometry was only performed for the novel compounds. To clarify this misunderstood, a sentence in the “Materials and Methods” was added. Meantime and as recommended, liquid chromatography studies were performed to evaluate bis-thiobarbiturates purity. A C18 column with DAD detector was used, and several elution systems were tested. Unfortunately, in the established seven days given to us by the editor-in-chef to present the manuscript revision, did not allow us to optimize the chromatographic process and to acquire other columns to successfully finish this task. Nevertheless, an excellent baseline on all NMR spectra seems to point to a notorious high purity of compounds where the presence of organic impurities seems to be negligible.

  • With regard to the in silico data, the study of authors is limited to pharmacokinetic aspects which, moreover, do not demonstrate that the samples are orally active as instead erroneously reported on page 12 line 25, but only provide information about bioavailability. The authors could do in silico studies of docking, which could improve the quality of the paper, given the existence of the crystal in the PDB database; otherwise, in my opinion, putting in silico studies in the title is misleading and therefore in my opinion it should be eliminated.

Thank you for this observation. The article title was changed accordingly. In addition, the text before Table 3 was modified to clarify this point and avoid the referred issue.

Reviewer 2 Report

Dear Editor,

Enclosed please find the comments on the following manuscript:

Manuscript ID: biomedicines-1400551

Title: Bis-thiobarbiturates as Promising Xanthine Oxidase Inhibitors: Synthesis, in vitro and in silico Studies.

In their manuscript, Serrano and coworkers synthesized sixteen bis-thiobarbiturates for screening the lead compound of xanthine oxidase inhibitors. Then, using structural bioinformatics approaches, they tried to rationalize the lead compound at molecular level.

  1. In my opinion, there are major concerns that impair the publication of this paper in the present form.
  2. The activity of these compounds has been tested without proper control. Indeed, the authors declared that they used a reference drug lopurinol (Allo). Authors should use febuxostat or topirastat as a positive control to gout drug discovery.
  3. Given that the authors performed antioxidant, anti-proliferative, and XO inhibitory IC50 activity without therapeutic index.
  4. The authors did not provide a solid molecular mechanism between antioxidant and XO inhibition.
  5. Table 1, Fig 1, Fig 2, and Fig 3: The resolution of chemical structures, numbers, and characters should be improved so that readers can read more clearly.

Author Response

Response to Reviewer #2 Comments

Manuscript ID biomedicines-1400551

Title: Bis-thiobarbiturates as Promising Xanthine Oxidase Inhibitors: Synthesis and Biological Evaluation

Dear Reviewer #2

The authors are extremely grateful and appreciative for the careful revision and very constructive comments made to the Biomedicines submitted manuscript and they would like to thanks for the opportunity to correct and enhance it. All the recommendations were taken into account and changes were made accordingly. Noteworthy main alterations and additions to the manuscript made in the manuscript are marked as word corrections. The responses to the questions and commentaries are herein answered. We hope that this final revised manuscript will be suitable for publication.

With kind regards, looking forward to your decision,

Paulo Almeida

In their manuscript, Serrano and coworkers synthesized sixteen bis-thiobarbiturates for screening the lead compound of xanthine oxidase inhibitors. Then, using structural bioinformatics approaches, they tried to rationalize the lead compound at molecular level.

The authors are very pleased and acknowledge the careful revision and the comments.

  1. In my opinion, there are major concerns that impair the publication of this paper in the present form.
  2. The activity of these compounds has been tested without proper control. Indeed, the authors declared that they used a reference drug lopurinol (Allo). Authors should use febuxostat or topirastat as a positive control to gout drug discovery.

The authors are grateful for this suggestion. Therefore, we performed the adequate experiments and, as suggested, Febuxostat was added as a positive control for the XO inhibition studies. Considering the experiments carried out and the observed results, some modifications in the abstract and throughout the remaining manuscript were made.

  1. Given that the authors performed antioxidant, anti-proliferative, and XO inhibitory IC50activity without therapeutic index.

The authors would like to thank the reviewer for this interesting suggestion. As in this manuscript we are presenting in vitro results for the development of a potential future lead compound, we considered that it was preferable to present selectivity index data. Therefore, attending to the fact that we have IC50 data for the cytotoxicity of compound 11 in non-tumoral cells, it was possible to present selectivity index values for this bis-thiobarbiturate concerning its effects as XO inhibitor and as radical scavenging agent. Accordingly, the adequate modifications were performed in the manuscript text in the paragraph after Figure 2.

  1. The authors did not provide a solid molecular mechanism between antioxidant and XO inhibition.

A scheme that represents the purine catabolism by XO with formation of reactive oxygen species was added to the SI (Scheme S2). The authors acknowledge this suggestion that undoubtedly led to an improvement in the paper quality and to a major information collected by readers.

  1. Table 1, Fig 1, Fig 2, and Fig 3: The resolution of chemical structures, numbers, and characters should be improved so that readers can read more clearly.

Changes were made accordingly and the resolution of all figures was improved.

Round 2

Reviewer 1 Report

Dear all,

I revised the paper submitted by Paulo Almeida and colleagues focuses on the synthesis of derivatives of barbiturates and the study of these derivatives as inhibitors of xanthine oxidase after their revision. In my opinion, the manuscript is ready for publication now.

Best regards

Reviewer 2 Report

In my opinion the revised manuscript is well written, the methodology and results clearly described and discussed. The research article would be interesting for drug discovery. I think that the manuscript is suitable for publication in the present form.